# Personality Traits, Personal Values, and Life Satisfaction among Polish Nurses

**DOI:** 10.3390/ijerph192013493

**Published:** 2022-10-18

**Authors:** Anna M. Zalewska, Marta Zwierzchowska

**Affiliations:** Institute of Psychology, SWPS University of Social Sciences and Humanities, Ul. Chodakowska 19/31, 03-815 Warsaw, Poland

**Keywords:** life satisfaction, ‘happy’ personality traits, ‘healthy’ values, trait–value consistency, value-environment congruence, nursing, integrated model of personality, psychological mediators, subjective well-being, eudaemonia

## Abstract

Nurses’ life satisfaction (LS) predicts their health and the level of care they provide to patients, thus policies for promoting quality of nurses’ work require actions to increase their LS. The aim of this study was to examine relations between LS and two levels of personality (traits and values) among Polish nurses, including joint effects of traits and values in a model integrating all variables to check whether meta-values can mediate trait–LS relationships. Nurses (*N* = 155) aged 23–64 completed the NEO-FFI, Satisfaction with Life Scale, and PVQ40. LS correlated with all traits, with openness higher than usual (0.34), and positively associated with meta-values: openness to change (0.23), self-transcendence (0.30), and (‘unhealthy’) conservation (0.19). Trait–value consistency was insufficient to explain some trait–value associations. In the SEM analysis, 23.3% of LS variance was explained. LS was related directly to neuroticism negatively and positively not only to extraversion, but also to openness, and self-transcendence meta-value (that increased value-environment congruence), and indirectly positively (through self-transcendence) to openness, agreeableness, conscientiousness, and even neuroticism. These results indicate that promoting nurses’ health and quality of work by enhancing their LS requires supporting and increasing their identification with self-transcendence values and encourage research on factors that can increase it.

## 1. Introduction

Subjective Well-being (SWB) refers to affective and cognitive evaluations of one’s own life according to subjective criteria [1,2]. Life satisfaction (LS) is a cognitive aspect of SWB that captures the overall sense of well-being from the one’s own perspective [2] and, depending on criteria, it can reflect hedonic (e.g., sensory pleasure) or eudaimonic (e.g., realization of one’s own potential) well-being [3,4,5]. In longitudinal research, life satisfaction was a predictor of future (reported in 6 years) depression symptoms [6]. There is some evidence that nurses have a relatively low subjective well-being, including LS (e.g., [7]), also in Poland [8]. However, the results of many studies show that lower levels of life satisfaction among nurses were significantly associated with higher levels of burnout [8,9,10], higher turnover rate (e.g., [11]) and lower level of care they provide to patients [12,13]. Moreover, higher satisfaction with life and work is related to greater readiness to learn and take on new professional challenges, and nurses with higher level of life satisfaction rated their preparedness for new competences as better [8]. Therefore, evaluation of life satisfaction and its determinants should be one of the main components of performing appropriate policies for promoting quality of work and improving productivity of nurses. Thus, it is important to identify factors that are responsible for nurses’ well-being. Previous studies on the general population indicate the importance of personality traits [14,15,16,17], as well as personal values for the level of life satisfaction [18,19,20]. However, it was common practice in these and most other studies to analyze LS relationships with traits or values separately. Few studies on LS have considered the role of both, traits and values (e.g., [21,22,23]). For example, Headey and Muffels [23] analyzed LS trajectories on the basis of data collected from 2500 respondents over 25 years (1990–2014) and concluded that LS stability is primarily due to stable personality traits, and medium and long term changes in LS are due to differences and changes in personal values/life priorities. Bojanowska and Urbańska [21] analyzed moderating effects of personality traits on relations between values and subjective well-being in national sample of Poland. In this study, we analyze relationships between all the considered variables (LS, traits, and values) in the professional group of nurses as there is a lack of such research on nurses. In addition, we also examine joint effects of personality traits and personal values on nurses’ life satisfaction, and determine whether values act as mediators of the trait-LS relationships, which has not been studied so far, referring to the integrative Model of Personality by McCrae and Costa [24].

### 1.1. Relations between Personality Traits and LS

In the Five-Factor Model (FFM), basic personality traits refer to biologically conditioned predispositions to think, feel, and behave that are relatively constant across time [24]. The model includes five basic traits (each defined by 6 facets): neuroticism (anxiety, angry hostility, depression, impulsiveness, vulnerability to stress, self-consciousness); extraversion (gregariousness, assertiveness, warmth, activity, excitement-seeking, positive emotions); openness to experience hereinafter interchangeably referred to as openness (fantasy, aesthetics, feelings, actions, ideas, values); agreeableness (trust, straightforwardness, altruism, compliance, modesty, tender-mindedness); conscientiousness (order, competence, dutifulness, achievement striving, self-discipline, deliberation).

McCrae and Costa [16] proposed that higher extraversion leads to more social and pleasure-related behaviors, which results in higher positive affect and higher LS, and higher neuroticism leads to higher negative affect and lower LS. Openness to experience predisposes individuals to feel negative and positive emotions more deeply and may help enlarge experience and realize own’s potential. Agreeableness enhances relationship quality, conscientiousness promotes the achievement of tasks, and they both may promote SWB by facilitating more positive experiences in social or achievement situations.

Results of meta-analyses confirm that all FFM traits are related to LS—neuroticism negatively and the others positively [14,15,17,25]. However, the data contained in these meta-analyses indicate that correlations between FFM traits and SWB constructs may vary by groups. In line with this observation, data collected in different age groups in Poland showed that Openness is more closely related to SWB among adults than adolescents [5], and among medical teachers in Pakistan the positive correlations of LS with agreeableness (0.64) and openness (0.39) were higher than usual [26]. Considering that LS of nurses is important for their work performance [12,13] and the satisfaction of patients under their care [27], there is a need to examine relationships between personality traits and life satisfaction among the professional group of nurses and check whether they are similar to those in the general population.

### 1.2. Relations between Personal Values and LS

Beside personality traits, another important determinant of life satisfaction are personal values [28]. Actions people take in pursuit of values have consequences for objective and subjective aspects of life. Research shows that one’s subjective well-being partly depends upon the set of values that one endorses [18]. For example, people for whom extrinsic and material values are especially important tend to have relatively low levels of subjective well-being [29].

One of the most influential psychological theories is the Schwartz theory of basic human values [30]. This framework defines values as stable and cross-situational cognitive representations of motivational goals that guide human behavior and serve as criteria for judging events [30]. Schwartz [30] distinguished ten basic values forming a motivational circle and grouped them into four higher-order meta-values. Two pairs of meta-values (1) self-transcendence versus self-enhancement and (2) openness to change versus conservation reflect motivational oppositions between competing values. The self-transcendence meta-value emphasizes concern for others, helping and care for people with whom one has frequent contact (benevolence) or expressing understanding, tolerance, and concern for all people and nature (universalism). The self-enhancement meta-value emphasizes the pursuit of self-interest by seeking social status and gaining control over people and resources (power) or by attaining personal success and demonstrating competence according to social standards (achievement). The openness to change meta-value emphasizes autonomy of thought and action (self-direction), enjoyment (hedonism), novelty, excitement, and challenge of life (stimulation). The conservation meta-value emphasizes commitment to past beliefs and customs (tradition), adherence to social norms and expectations (conformity) and preference for stability and security for self and others close to self (security).

However, more recent publications (e.g., [20,31]) emphasize the importance of two other principles organizing the value circle—primary goals expressed in values and the relation of values to anxiety. Person-focused values (openness to change and self-enhancement) regulate goals expressing personal interests and social-focused meta-values (self-transcendence and conservation) relate to social interactions and cooperation. Self-transcendence and openness to change are relatively anxiety-free and represent self-expansion/growth values called as ‘healthy’ [18]. Self-protection and anxiety-control or deficiency values (called ‘unhealthy’) include conservation, power and part of achievement (controlling anxiety by meeting social standards). However, achievement also has a growth component (affirming the sense of competence).

Sagiv and Schwartz [18] assumed that SWB (LS and positive affect) is positively correlated with ‘healthy’ values and negatively with ‘unhealthy’ values. However, empirical data did not confirm this pattern as five values (universalism, achievements, tradition, security, and conformity) showed no clear trends [18,32]. Moreover, research on university students from 14 nations showed that Life satisfaction was predicted by self-transcendence and conservation positively, and by interaction of self-directed hedonism with self-enhancement [33].

The data collected so far indicate that the values-LS relationships are not universal, as they are moderated by socio-cultural factors—by Human Development Index—HDI [32], HDI and cultural egalitarianism (high equality, tolerance, loyalty, help and care for others [20]), and many other factors [19]. Some of them, especially value-environment congruence [19] and contextual threats [34] may concern the professional group of nurses.

Poland reached the average level of HDI and cultural egalitarianism among 32 European countries participating in the study of Sortheix and Schwarz [20]. Thus, we can expect that openness to change value, which motivates to satisfy intrinsic needs (autonomy, relatedness and competence—Ryan and Deci [35]), will promote LS among Polish nurses as among people in different countries (e.g., [20]).

The working conditions of nurses involve many unpredictable, threatening, and anxiety-arousing events because of contact with the suffering (and death) of patients, which directly and indirectly (e.g., by confronting one’s own mortality) influences the level of stress and threats in work environment [36]. It can also rise ethical conflicts regarding nurse role and relationships with co-workers, patients, and their families [37]. Therefore, personal values are of particular importance in the nursing profession, helping nurses to manage stressful situations at work and to resolve ethical conflicts they are exposed to [37]. In the face of different contextual threats (e.g., migration [38]; global financial crisis [39]), identification with self-enhancement or openness to change (person-focused) values decreased and social-focused (self-transcendence and conservation) values increased. Boer [34] also found that contextual threats foster links between affective well-being and social-focused values that protect against these threats. Thus, we assume that among nurses social-focused values will be more important than person-focused, and by protecting nurses from stress and threats in work environment, they will be positively related to LS.

For most Polish nurses, the self-transcendence value (‘the desire to help other people’) is the main reason for choosing a profession [40]. It is the core value of the nursing profession. Higher identification with it may help nurses maintain a definition (meaning) of work as helping other people and achieve valued goals (daily care of patients), increasing their level of job satisfaction [40], which, in turn, is related to higher LS (e.g., [41]). Higher identification with this value will increase value-environment congruence, which, by increasing cooperation in achieving valued goals, social support, and reducing internal value conflicts, also promotes LS [19]. We therefore expect self-transcendence to be most positively related to nurses’ LS.

Identification with conservation value involves adherence to social norms and acceptance of orders and prohibitions. Higher adherence to medical recommendation can be crucial in effective patient care, leading to higher LS. Moreover, it may be adaptive in threatening and anxiety-arousing environments [38,39] because it can increase the sense of security [34], which is especially important in the context of contact with the suffering (and death) of patients and the threat to one’s health.

The self-enhancement value is associated with seeking social status and gaining control over people and resources or attaining personal success and social approval. Striving to achieve such goals could be difficult in the working environment of nurses and would therefore be related to lower LS. However, some degree of control over patients is inherent to the occupational role of nurses, and achievements can rely on developing one’s competence as a core intrinsic goal [35]. Thus, it would be related to higher LS. However, the inconsistent theoretical data and the lack of research on the relationship between self-enhancement and LS among nurses leave an open question about the relationship between personal values and life satisfaction among nurses.

### 1.3. Personality Traits and Personal Values, and Joint Effects of Traits and Values on LS

As shown above, the studies conducted so far in general populations confirm the associations between life satisfaction and these two variables, traits and values, understood as separate constructs. Responding to calls for the integration of personality research [42], especially in the field of subjective well-being [43], and for research on common effects of traits and values [19] we use an additional approach in this study. Referring to a more integrated model of personality [24] that goes beyond the classic trait only approach [42] we consider relationships between nurses’ LS and traits and values as variables representing two levels of personality. The first level is represented by traits described in the FFM [24], and the second is represented by meta-values [31] because ‘values are characteristic adaptations shaped by the interplay of basic personality traits and the stimuli of the social and natural world’ [44] (p. 45).

Traits and values are considered as distinct components of personality. Traits are basic biologically conditioned and relatively stable tendencies in thinking, feeling, and behavior [24]. Values are cognitive transformations of needs or drives into conscious goals resulting from biological needs, social interactions and group requirements [30]. Although they have similar genetic foundations [45], values are dependent on social factors to a higher degree [46]. Moreover, traits describe what people are like (how they behave) and values indicate what they engage in—what is important to them [47].

However, they are interrelated [22,46,47,48]. Hypotheses about trait–value relations (e.g., [47,48]) are primarily based on the similarity in the content (consistency) of particular traits and values. For example, people high in agreeableness tend to help others and cooperate with them, so they probably value: understanding, tolerance, and concern for all people-universalism and benevolence-and respect for customs and social norms-tradition and conformity. In their meta-analysis, Parks-Leduc et al. [48] confirmed previous findings (e.g., [47]). They reported that openness positively correlates with targeting, stimulation, and universalism. Agreeableness is positively associated with benevolence, universalism, conformism, and tradition. Extraversion has positive relations with stimulation, hedonism, power, and achievements. Conscientiousness positively correlates with safety, conformism, and achievements. They showed also negative relationships indicating that openness negatively correlates with tradition, conformism, and security (conservation), and agreeableness—with power and achievement. They did not find consistent relations between neuroticism and values.

Since there is a lack of such research on nurses there is a need to examine relationships between personality traits and personal values in the professional group of nurses and check whether the assumption about trait–value consistency (e.g., [47,48]) can explain these relations.

To examine joint effects of traits and values [19] we assume that traits work in conjunction with values contributing to well-being. Basic traits are relatively stable and they co-determine with external factors characteristic adaptations, including values, and through these adaptations indirectly influence behavior [24] and LS [43]. This mediation model, resulting from theoretical considerations [24,44] is also justified by the results of longitudinal studies, which showed that traits predicted values more strongly than they were predicted by values [22]. Values are conscious cognitive representations of motivational tendencies shaped in the life course by the interplay of personality traits and sociocultural factors [44]. They motivate for attaining valued goals and also serve as criteria for judging events [30,31] directly influencing LS. We assume that achieving important goals defined by the values and consistent with traits underlying them will promote LS. Therefore, we suppose that values, which are related to LS, can mediate the relationships between the traits consistent with them and LS.

### 1.4. Aim of the Study and the Hypotheses

The overall aim of this study is to investigate relationships between FFM traits, personal values and LS among nurses. This aim has two specific objectives: (1) to examine simple associations between all the considered variables as separate constructs like in earlier research (traits and LS, traits and values, and values and LS) in the professional group of nurses as there is a lack of such research on nurses; and (2) to examine these relationships in a complex model integrating all variables in order to determine common effects and check whether meta-values mediate traits-LS relationships, which has not been studied so far.

Based on theoretical and empirical premises presented above we have formulated 4 hypotheses:

**Hypothesis** **1** **(H1).**
*Life satisfaction of Polish nurses will be negatively related to neuroticism and positively to extraversion, agreeableness, openness to experience, and conscientiousness.*


**Hypothesis** **2** **(H2).**
*Identification with openness to change, self-transcendence, and conservation (growth or social-focused) values will be positively related to the LS of Polish nurses and the self-transcendence-LS relationship will be the most positive.*


**Hypothesis** **3** **(H3).**
*Extraversion will be positively related to self-enhancement and openness to change values (person-focused). Agreeableness will be positively related to self-transcendence and conservation (social-focused), and negatively related to self-enhancement. Openness will be positively related to openness to change and self-transcendence (growth values), and negatively related to conservation. Conscientiousness will be positively related to self-enhancement and conservation (self-protection values).*


**Hypothesis** **4** **(H4).***Personal**meta-values—openness to change, self-transcendence, and conservation—**will mediate the relationships between FFM traits consistent with them and LS among Polish nurses: identification with openness to change value can be a mediator for traits openness and extraversion, with self-transcendence value—for traits agreeableness and openness, and identification with conservation value—for traits conscientiousness and agreeableness*.

## 2. Methods

### 2.1. Participants and Procedure

Taking care of the representativeness of the sample in terms of the length of patient care, patient age, type of care provided by nurses and patient mortality, the study included 155 female nurses aged 23–64 (*M* = 44.6, *SD* = 9.58) working in psychiatric (*N* = 46), oncology (*N* = 52) and midwifery (*N* = 57) departments. Workplace type was linked to nurses’ age—*F* (2, 152) = 5.83, *p* = 0.004. The psychiatric nurses were older (*M* = 48.3, *SD* = 8.91) than oncology nurses (*M* = 44.1, *SD* = 9.01, *p* = 0.067) and midwives (*M* = 42.0, *SD* = 9.74, *p* = 0.003). Regardless of their specialization, all nurses underwent the same education process, and hence, we can infer that the age of nurses corresponds to their working experience.

In 2015, the questionnaire sets were sent to 354 female nurses, as in 2015 98.3% of people registered as nurses in Poland were women [49]. The sets were distributed in nurses’ workplace for self-completion. Each set contained contact details of the researcher enabling contact between respondents and researcher, information about the purpose of the study, the way the collected data will be used and a stamped and addressed return envelope for completed questionnaires. The return rate was only 50.2%, and after excluding 17 questionnaire responses with errors and missing data, we analyzed data from 155 nurses.

We followed all ethical standards in accordance with The Code of Ethics of the World Medical Association (Declaration of Helsinki) and the study was approved by the Ethics Committee for Scientific Research at SWPS University of Social Sciences and Humanities (resolution 1/2015). We received informed consent from all participants.

### 2.2. Measures

#### 2.2.1. Personality Traits

FFM traits were assessed using the Polish adaptation of the NEO-FFI [50]. The questionnaire consists of 60 items (12 for each trait-2 for each facet) with a 5-point scale (from ‘I completely disagree’ to ‘I completely agree’). Example items are as follows: Neuroticism, ‘I am not someone who constantly worries’; for Extraversion, ‘I like to have many people around me’; for Openness, ‘I don’t like wasting time on dreams’; for Agreeableness, ‘I try to be polite to everyone I meet’; and for Conscientiousness, ‘I keep my things clean and tidy’. The reliability (Cronbach’s alpha) of the Polish version is sufficient (from 0.68 to 0.82).

#### 2.2.2. Life Satisfaction

LS was measured using the Satisfaction with Life Scale (SWLS [1]). The Polish version by Juczyński [51] has high reliability (α = 0.82). Participants indicate to what extent they agree with the five statements about their lives (e.g., ‘In most ways, my life is close to my ideal’) on a scale from 1 (‘I definitely disagree’) to 7 (‘I definitely agree’). Higher scores indicate higher LS.

#### 2.2.3. Personal Values

The basic 10 values, grouped into four meta-values, were estimated with the Polish adaptation [52] of the Portrait Values Questionnaire—PVQ40 developed by Schwartz, which consists of 40 items. Each item describes what is important to different women. For each item (e.g., ‘It is very important for her to help other people around her. She wants to care for her personal well-being’), respondents answer the question ‘How much like you is this person?’ on a 6-point scale (from 1—‘completely unlike me’ to 6—‘very similar to me’). The reliability of the ten subscales and four meta-values was sufficient (Cronbach’s alpha: 0.60–0.70 and 0.75–0.79, respectively).

### 2.3. Statistical Analyses

First, we tested hypotheses H1–H3 about simple relations between variables as separate associations (like in earlier research) using Pearson correlations coefficients. Finally, to examine the relationships while controlling for the effects of all variables and to determine joint effect and check whether meta-values mediate traits-LS relationships, we conducted structural equation modelling (SEM) analysis using AMOS 26 graphics. Before the main analyses we conducted preliminary analyses (Harman’s single factor test, internal consistency analyses, skewness and kurtosis, multicollinearity indices) using IBM SPSS Statistics 26.0 for Windows to check if the assumptions are met.

## 3. Results

### 3.1. The Results of Preliminary Analyses

The result of Harman’s single factor test for all items (12.3% < 50%) showed that there was no problem with a common method bias. The Cronbach’s alpha indices for nine variables (from 0.68 to 0.87) were satisfactory. For the trait Openness, we removed one item, which in the examined group correlated negatively with the whole scale and calculated the index of Openness on the basis of 11 items (by multiplying the mean for 11 items by 12). Cronbach’s alpha for such scale was acceptable (0.59) for group analyses [53]. The skewness and kurtosis values did not exceed the absolute value of 1 for each variable, so we can use parametric tests [53]. Finally, the Variance Inflation Factor (VIF) indices were lower than 3 (1.42–1.80 for traits and 1.84–2.54 for meta-values), showing that there was no issue with the high multicollinearity of predictors.

We also checked differences in the level of nurse identification with 4 meta-values (value importance). Results of the six *t*-tests for dependent group showed that importance of four meta-values varied from one another (*t*(154) values ranged from 7.62 to 15.55, *p* < 0.001; effect sizes ranged from medium 0.57 to large 1.76). Identification of nurses with self-transcendence dominated, next was conservation and then openness to change, with the least important for them self-enhancement (see Table 1). These results support our assumption that due to contextual threats among nurses, social-focused values are more important to them than person-focused and that their higher identification with the self-transcendence fosters higher value-environment congruence.

### 3.2. Descriptive Statistics and Correlations between Variables in the Whole Group

In Table 1, basic descriptive statistics, reliability indices and intercorrelations between all variables, including age, are presented. All variables were not significantly related to age, therefore age was omitted in further analyses.

According to Cohen [54], correlation values of 0.1 ≤ r < 0.3 should be considered small effects, values of 0.3 ≤ r < 0.5—medium, and values of r ≥ 0.5 should be considered large effects. Based on these guidelines, higher scores on extraversion, openness, conscientiousness, and agreeableness were related to higher LS (medium to small effects), while higher scores on neuroticism were related to lower LS (medium effect). Higher LS scores were related to higher scores for the meta-values: self-transcendence (medium effect), openness to change, and conservation (small effects). Only self-enhancement was not related to LS. These findings fully confirmed hypotheses H1 and H2. Moreover, we found 12 significant relations between traits and values—8 (of 10) expected relations in H3 and 4 additional relations. As expected, agreeableness was positively related to self-transcendence and conservation (social-focused) and negatively related to self-enhancement. Extraversion was positively related to self-enhancement and openness to change (person-focused) and additionally related to self-transcendence. Conscientiousness was positively related to conservation; however, its relationship with self-enhancement was not significant, and it was positively related to openness to change and self-transcendence (growth values). Openness was positively related to openness to change and self-transcendence values (growth values), but it was not related negatively to conservation. Finally, neuroticism was negatively related to openness to change (small effect).

The values were positively related to each other, except that self-enhancement was not related to conservation and self-transcendence (social-focused). Openness to change was positively correlated with all meta-values, most strongly (0.62) with self-enhancement (person-focused). The strongest effect (0.71) was found for the relationship between self-transcendence and conservation (social-focused). The data in Table 1 also show that the FFM traits were intercorrelated. Neuroticism was negatively related to all traits (medium to strong effects), and the relationships between all other traits were positive (small to medium effects).

### 3.3. Complex Relations between Traits, Meta-Values and Life Satisfaction in SEM

Finally, using SEM analysis, we tested the complex relationships between all examined variables while controlling for all their effects. We created an initial model with LS and four meta-values (conservation, self-transcendence, openness to change and self-enhancement) as five observed endogenous variables, five unobserved exogenous variables (e1 to e5), and the five FFM traits (N, E, O, A, C) as observed exogenous variables. We also included the intercorrelations between personality traits and between four unobserved exogeneous variables (e1 to e4—representing the residual terms of the meta-values), in the model. This analysis allowed us to determine all direct effects of the FFM traits on the meta-values (H3), all direct effects of the traits (H1) and meta-values on LS (H2), as well as all the indirect effects of the traits on LS mediated by meta-values (H4). After drawing the initial model, we computed estimates, removed nonsignificant paths and analyzed the fit indices. The model fit was evaluated based on goodness-of-fit indices provided by the software and discussed by Kenny [55]: χ^2^ [*df* = 19, *N* = 155] = 26.44, *p* = 0.118, χ^2^/*df* = 1.392, Tucker–Lewis index (TLI = 0.962), comparative fit index (CFI = 0.984) and root mean square error of approximation (RMSEA = 0.050). All the indices indicated good fit in reference to the cut-off points (χ^2^/*df* ratio of 3 or less, TLI, CFI > 0.95, RMSEA < 0.06 [55]).

Figure 1 presents the regression weights of the estimates of the model paths. We found that each meta-value was directly related to a different set of traits. We identified 8 of 10 expected (H3) direct relations, as in correlational analyses, and 2 additional predictors. Person-focused values—openness to change and self-enhancement—were positively predicted by extraversion. Openness to change was also positively linked to openness and self-enhancement was negatively related to agreeableness. The traits explained ca. 6.5% of the variance in self-enhancement and 22% of the variance in openness to change. Social-focused values—conservation and self-transcendence—were positively predicted by agreeableness and conscientiousness. Self-transcendence was also positively related to openness and neuroticism (though its simple correlation with neuroticism was not significant—see Table 1). The traits explained ca. 24% of its variance and ca. 9% of the conservation variance.

In the complex model, ca. 23.3% of LS variance was explained. LS was directly related to neuroticism (negatively) and positively related to extraversion and openness (what partially confirms H1), as well as to the meta-value self-transcendence (what partially confirms H2). Moreover, we also found positive indirect effects of the four FFM traits on LS mediated by self-transcendence value (inconsistent with H4). The standardized indirect effects of traits were as follows: β = 0.028 for neuroticism, β = 0.043 for openness, β = 0.055 for agreeableness, and β = 0.058 for conscientiousness. The other meta-values were not related directly to LS and did not mediate the relationships between the FFM traits and LS.

## 4. Discussion

In this study, we examined the relationships between FFM traits, personal values, and LS among Polish nurses, because their LS level is related to their health and quality of work. It can predict future depression symptoms [6], it is related to the level of burnout (e.g., [8,9,10]), the level of care they provide to patients [12,13], and the turnover rate of nurses [11]. We investigated these relationships to check whether they are similar to those observed in the general population. We tested three hypotheses regarding basic relations using simple correlations like in earlier research. However, responding to calls for the integration of personality research [42], especially regarding SWB [43], and for research on joint effects of traits and values [19] we also tested them in a complex model integrating all variables, checking whether meta-values mediated traits-LS relationships.

### 4.1. Correlations between Personality Traits and Life Satisfaction

The results of zero-order correlations fully confirm the hypothesis H1. Similar to findings from other studies (e.g., [14,15,16,17] Polish nurses’ LS was related to all FFM traits, negatively to neuroticism and positively to the others, but its correlation with openness to experience was higher than usual (e.g., [14]) like among medical teachers in Pakistan [26]. This result will be discussed further.

### 4.2. Correlations between Personal Values and Life Satisfaction

Consistent with hypothesis H2, Polish nurses’ LS positively correlated with three meta-values-openness to change, self-transcendence, and conservation-and the self-transcendence-LS relationship was the most positive (medium effect). The results confirm theoretical assumptions and earlier data [20,32] that openness to change (growth and person-focused) value promotes LS. This value may help nurses engage in preferred by them activities and satisfy their intrinsic needs. Contrary to the claim that social-focused values are not beneficial to well-being as they cause concentration on others [20], both these values promoted nurses’ LS. The findings are in line with our assumption that the work environment of nurses is anxiety-arousing, therefore social-focused values are more important to them than person-focused (Table 1), they protect nurses against stress and threats, and positively correlate with LS [34]. Like the results from the national Polish sample where LS was positively related to all meta-values [21], they question the hypotheses that conservation (self-protection and social-focused) value is ‘unhealthy’ [18] and commonly undermines LS [20]. However, in the national sample the self-transcendence-LS relationship was very weak [21], while among Polish nurses it was the most positive, as expected, assuming that the higher identification of nurses with this fundamental value for a profession [40] increases value-environment congruence and its positive impact on LS [19]. These results can also mean that Polish nurses professional subculture due to this core value may be more egalitarian [20] than the culture of the country of Poland.

### 4.3. Correlations between Personality Traits and Personal Values

Nurses’ meta-values were related to specific sets of traits. The relations partially confirmed the hypothesis H3, based on the consistency (similarity in the content) of these constructs [47,48]: extraversion was positively related to both person-focused values, openness to both growth values, agreeableness to both social-focused values, and additionally it was related negatively to self-enhancement value.

Contrary to the hypothesis H3, conscientiousness positively correlated with both social-focused values and openness to change, as in the meta-analysis by Fisher and Boer [46]. Moreover, it was not related to self-enhancement, and openness was not negatively related to conservation value. Similar results have recently been obtained in a national sample representative of gender, age, education and the regions of Poland [21]. Poland is a country ‘between individualism and collectivism’ in which individualism and conservative values are accepted by a large part of Polish society [5,56]. Perhaps such a ‘mixed’ cultural context may modify motivational oppositions and some trait–value links [46]. Therefore, openness to change is positively related to all other meta-values (Table 1) and openness trait does not collide with conservation meta-value, but may mean tolerance for it, and conscientiousness is related to all meta-values except self-enhancement that is most similar in content.

### 4.4. The Relationships between FFM Traits, Meta-Values and LS in the Complex Model

In the complex model (SEM analysis), the LS of Polish nurses was directly predicted negatively by neuroticism, and positively by extraversion, openness (H1) and self-transcendence meta-value (H2). These results confirm earlier findings that neuroticism and extraversion determine ‘happy’ personality and play important role in predicting LS (e.g., [14,15]). However, they indicate that not only extraversion, but also openness was ‘happy’ trait and positively predicted the nurses’ LS.

A high level of extraversion is associated with positive affectivity and activity-being socially active, optimistic, talkative and assertive [57]. Extraverted nurses may therefore rate life events, activities, and relationships with others, including patients and co-workers, as less stressful, more enjoyable, and fulfilling than introverted ones.

Openness involves creativity and a willingness to learn new things [58], so higher openness helps to better cope with unpredictable working conditions of nurses. Higher openness is associated with better social adaptation and understanding others [58], better conflict management and social support [59], and better quality of interpersonal relations [60]. All these characteristics appear to be important for professional patient care, they can contribute to broadening of the nurses’ experience and realization of their own potential in a professional role, and also result in a higher LS. Such a dependence that openness directly predicts nurses’ LS may partially explain the data that a higher LS of nurses is associated with greater readiness to learn and take up new professional challenges, and a better assessment of their own preparation for new duties [8].

In the model integrating all variables, only self-transcendence value predicted LS. The self-transcendence motivates to help and take care of others. Higher identification with it as the core value of the nursing profession [40] increases value-environment congruence. Better congruence allows nurses to attain important goals (to express concern for others), to receive social support, and to avoid or resolve inner conflicts [19,37], leading to higher LS. Additionally, expressing this value in everyday life protects nurses against stress and threats, can help them control anxiety, and thus protects their affective well-being [34], as well as LS.

The results did not confirm the hypothesis H4. Although openness to change and conservation meta-values positively correlated with nurses’ LS, they did not predict it when effects of all variables were controlled and thus could not mediate trait effects. The correlation between openness to change and LS disappeared, presumably because it was the result of the influence of extraversion and openness on both variables—value and LS. Thus, the impact of openness to change on LS was included into these traits effects. The conservation effect most likely was included into the self-transcendence effect on LS, as conservation and self-transcendence were strongly correlated. Moreover, self-transcendence meta-value did not mediate the relationships between its compatible traits and LS, as we found positive indirect effects of the four traits (except extraversion) on LS via self-transcendence value. One of them was openness responsible for tolerance and the size of social network [59], openness to ideas, behaviors, and emotions of oneself and others, and understanding others [58]. Additionally, agreeableness and conscientiousness, which represent Socialization [61], the meta-factor of personality responsible for smooth social relations and willingness to cooperate, positively predicted LS through self-transcendence. Moreover, the insignificant simple correlation of neuroticism with self-transcendence became significant positive relationship, and higher neuroticism through higher self-transcendence positively predicted LS when the suppressive effects of other traits were controlled (Figure 1).

Neuroticism determines sensitivity to threats and aversive stimuli [24], and susceptibility to intolerance of uncertainty, when the negative response to unknown events is even stronger than to negative ones [62]. Neuroticism directly negatively predicted nurses’ LS. However, by enhancing the experiences of threats and uncertainty at work, higher neuroticism increased the nurses’ identification with self-transcendence value, as do contextual threats [38,39]. Higher self-transcendence value allows nurses to focus on helping others, thereby reducing the negative response to unpredictability and threats [34], protecting and even increasing their level of LS. The mediating effect of self-transcendence value in the relationship between neuroticism and LS is an important finding of our study that encourages further research in other groups of nurses and other professional groups who help others.

Bojanowska and Urbańska [21] found that openness, agreeableness and conscientiousness are more closely related to eudaimonic than subjective well-being. In this study, the role of openness in predicting the nurses’ LS was higher than usual. Moreover, the four traits (except extraversion) indirectly predicted LS through the self-transcendence value that is fundamental for the nursing profession [40]. Considering that LS may reflect hedonic and eudaimonic well-being [4,5] these results suggest that Polish nurses’ LS may reflect eudaimonic well-being (commitment to goals that give meaning to their work and life) more than LS in the general Polish population.

The nurses’ LS predicts their health and quality of their work, especially the level of care they provide to their patients, thus the policies of promoting the quality of nurses’ work require measures to increase their LS. The results of this study prove that FFM personality traits and self-transcendence values are important predictors of nurses’ LS. Given that FFM traits are relatively stable and personal values can be altered to a greater extent [22,63], our results indicate that promoting health and quality of work of nurses by increasing their LS requires interventions that can support or strengthen their identification with self-transcendence values.

### 4.5. Limitations

This study has several limitations. The data were gathered by self-reports only. Correlational nature of the study does not allow us to conclude about cause-and-effect relationships. The statistical analyses used only consider linear relationships and are not resistant to outliers. The sample was collected from only three departments. The study was based on a convenience sample, and therefore, the results and conclusions may be limited. The sample included Polish nurses living in a country with both collectivist and individualist cultural elements and specific socioeconomic conditions; the results may vary in samples with other cultural and socioeconomic characteristics. Finally, other variables that were not included in this study may affect explained variables.

## 5. Conclusions and Future Directions

Despite its limitations, the current study casts new light on the relationships between FFM traits, values, and LS, because it investigated them among a specific professional group of nurses and examined the relationships not only as separate correlations, but also as joint effects of traits and values in a complex model integrating all variables.

The results constitute another voice in the debate on ‘healthy’ values [18,19,20], proving that openness to change, self-transcendence, and also conservation values promote the nurses’ LS. In line with Fisher and Boer’s [46] findings, they indicate that trait–value consistency (in content) is not sufficient to explain some trait–value associations among Polish nurses, as they probably depend on contextual factors-Polish ‘mixture’ of individualism and collectivism and specific professional subculture of nurses.

As the first study to show that the self-transcendence value as fundamental for nursing profession mediates positive indirect effects of the four FFM traits (excluding extraversion, but including neuroticism) on LS among Polish nurses we filled the gap and provided deeper insight into the examined relationships than ‘zero-order’ correlations. These results, as well as the higher than usual role of all traits, especially openness in predicting LS suggest that Polish nurses’ LS may be more saturated with eudaimonic well-being than in general population. They encourage further research into other groups of nurses (also in various countries) and other professional groups who help other people.

While FFM traits are relatively stable, personal values can be changed to some extent [22,63]. Our results may be helpful for institutions employing nurses, pointing to the importance of institutional interventions aimed at nurses, supporting their identification with self-transcendence values. In this way, employers may try to protect the life satisfaction of nurses, while contributing to the improvement of the quality of their work and indirectly contributing to the increase in patient satisfaction with care. The obtained data also indicate a field for individual therapeutic interventions aiming to increasing identification with personal values, especially with self-transcendence values, and thus protecting the life satisfaction of nurses. This may be the focus of further research on the various factors and interventions that can increase the identification with self-transcendence among nurses to improve their life satisfaction, and as a result to promote their health and quality of work.

## Figures and Tables

**Figure 1 ijerph-19-13493-f001:**
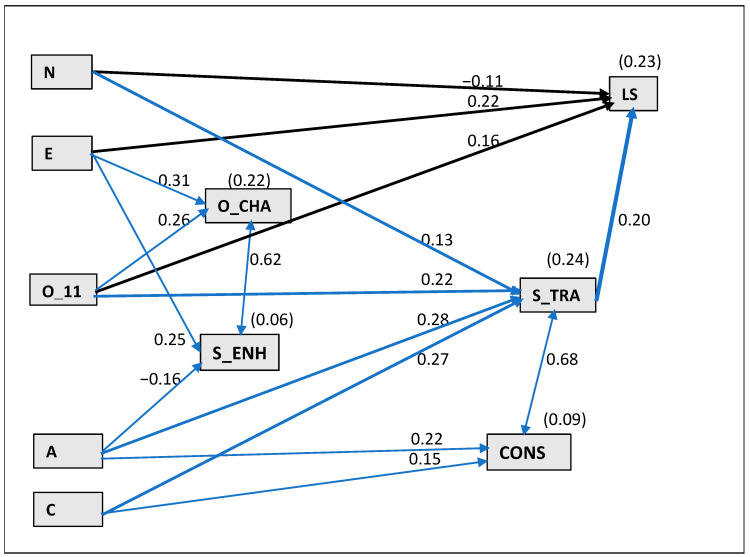
The model of relationships between FFM traits, meta-values and life satisfaction with standardized path coefficients. Note: FFM traits: N—Neuroticism, E—Extraversion, O_11—Openness to experience (based on 11 items), A—Agreeableness, C—Conscientiousness; Meta-values: O_CHA—Openness to change, S_ENH—Self-enhancement, CONS—Conservation, S_TRA—Self-transcendence; LS—Life satisfaction.

**Table 1 ijerph-19-13493-t001:** Means, standard deviations, and reliability (Cronbach’s α) of the variables and the intercorrelations between traits, meta-values, LS, and age (*N* = 155).

	*M*	*SD*	Alpha	1	2	3	4	5	6	7	8	9	10
1 N	21.93	7.00	0.79										
2 E	26.97	5.40	0.68	−0.54 ***									
3 O_11	25.62	5.05	0.59	−0.35 ***	0.36 ***								
4 A	30.77	5.04	0.68	−0.35 ***	0.31 ***	0.26 **							
5 C	33.23	6.20	0.83	−0.50 ***	0.45 ***	0.23 **	0.28 ***						
6 O_CHA	3.46	0.74	0.75	−0.25 **	0.40 ***	0.40 ***	−0.06	0.28 **					
7 S_ENH	3.02	0.87	0.79	−0.15	0.20 *	0.11	−0.16 *	0.13	0.62 ***				
8 CONS	4.05	0.73	0.78	−0.03	0.10	0.07	0.26 **	0.22 **	0.20 *	0.10			
9 S_TRA	4.49	0.78	0.79	−0.10	0.18 *	0.29 ***	0.36 ***	0.33 ***	0.24 **	0.002	0.71 ***		
10 LS	20.26	6.16	0.87	−0.31 ***	0.37 ***	0.34 ***	0.16 *	0.26 **	0.23 **	−0.01	0.19 *	0.30 ***	
11 Age	44.55	9.58	---	−0.06	−0.09	0.08	0.12	−0.03	−0.13	−0.04	0.08	0.04	−0.12

Note: FFM traits: N—Neuroticism, E—Extraversion, O_11—Openness to experience (based on 11 items), A—Agreeableness, C—Conscientiousness; Meta-values: O_CHA—Openness to change, S_ENH—Self-enhancement, CONS—Conservation, S_TRA—Self-transcendence; LS—Life satisfaction. * *p* < 0.05; ** *p* < 0.01; *** *p* < 0.001.

## Data Availability

The data that support the findings of this study are available from the corresponding author upon reasonable request.

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
