# Peer review of "Personality Traits, Personal Values, and Life Satisfaction among Polish Nurses"

_ijerph, 2022, doi:10.3390/ijerph192013493_

Round 1

Reviewer 1 Report

I am grateful to the editorial board for having been chosen to review this work. It is clear that the authors have worked hard on it. The following are the comments

Abstract

It is necessary to add a sentence related to the background or justification of the objective.

It is also important to add brief information about the type of analysis in order to understand the results and to add quantitative data to the results.

Modify the sentence "new insight into the examined relationships" and make it more specific.

Introduction

The need for the work is not clear. That is, there is only one sentence at the beginning of the introduction stating the importance of identifying factors that are responsible for nurses' well-being with a citation [1] that is from 2015 and is not a meta-analysis (27-30). They should add more literature that relates this to the objective.

Then it is stated that there are "few studies" but none of them are mentioned (30).

The introduction is for the justification of the work, but in this section the procedure is not indicated. The procedure would go in the "methodology" section (42-46), and also the hypotheses that the authors have included in the introduction (68-71; 120-127;179-184; 196-205), should all go after the objectives. So, at the end of section 1.4, they should summarise what is indicated in the introduction to include the Aim of the study and the hypotheses. This should be in scientific format.

In addition, the different sections of the introduction have to be better connected to each other.

Methods

The procedure section lacks information on how the questionnaire was sent to them, whether there was someone supervising, whether they were able to resolve doubts during the completion of the questionnaire, the year in which the information was collected, etc.

The section on statistical analysis is missing, where the type of analysis is reported and justified. The authors should indicate which analysis was carried out to resolve each hypothesis put forward.

Conclusión

They need to include real practical implications of their work. It is noticeable here that, as there is no good justification of the need for their study in the introduction, the implications are not clearly reflected in the conclusion either. Therefore, beyond the need for further study, they need to include information on the clinical implications of their work.

Rerences

Most of the references they use are very old. More current references should be used in the introduction and in the discussion and conclusion.

It is necessary to improve.

Reviewer 2 Report

This is a study of great quality and scientific value that deserves to be published. However, the following questions should first be addressed:

- It is necessary to add Personality traits and life satisfaction in the Keywords.

- Nurses’ well-being is not a "little studied topic" as stated in the first paragraph of the introduction (there are more than 46,000 results since 2018 in Google Scholar) and such a statement cannot be made without support in other references or explored throughout several paragraphs the state of the matter, so it is essential to improve and expand it considerably.

- It would be advisable to include the Gender variable when describing the sample, as well as the type of sampling, the criteria for selecting the sample as representative of the population, and, if the gender is known, it would also be advisable to include a population pyramid.

My congratulations and a cordial greeting to the authors.
